# Effects of a Nanonetwork-Structured Soil Conditioner on Microbial Community Structure

**DOI:** 10.3390/biology12050668

**Published:** 2023-04-28

**Authors:** Jingjing Cao, Pan Zhao, Dongfang Wang, Yonglong Zhao, Zhiqin Wang, Naiqin Zhong

**Affiliations:** 1State Key Laboratory of Plant Genomics, Institute of Microbiology, Chinese Academy of Sciences, Beijing 100101, China; caojingjing@im.ac.cn (J.C.); ylzhao97101@163.com (Y.Z.); wangzq1210@163.com (Z.W.); 2Engineering Laboratory for Advanced Microbial Technology of Agriculture, Chinese Academy of Sciences, Beijing 100101, China; 3The Enterprise Key Laboratory of Advanced Technology for Potato Fertilizer and Pesticide, Hulunbuir 021000, China; 4College of Environmental Science and Engineering, Donghua University, Shanghai 201620, China; dfwang@dhu.edu.cn

**Keywords:** nanocomposite microbial community, control nitrogen loss, soil properties, high-throughput sequencing

## Abstract

**Simple Summary:**

Soil is the basis for providing “fertile ground” for agriculture, including the growth of food and bioenergy crops. The nutrients needed by plants can be provided by chemical fertilizers. However, the heavy usage of fertilizers not only restricts crop production but also destroys the soil environment. Meanwhile, the soil microbiome, an indicator of soil quality that can affect plant growth and yield production, also changes. To solve this problem, a network-structured nanocomposite was prepared for use as a soil conditioner (SC). It was shown that this SC has a good ability to control nutrient loss; advance the agronomic traits of pepper, including growth and yield; improve soil physicochemical properties; and facilitate changes to the composition of microbial communities. With the view that the soil microbiome could develop soil function, we analyzed the correlation between network-structured nanocomposites and soil, microorganisms, and plants. In conclusion, nanonetwork-structured SC is an ideal technology that is effective for the promotion of crop growth and the improvement of the soil microbial community.

**Abstract:**

Fertilizer application can increase yields, but nutrient runoff may cause environmental pollution and affect soil quality. A network-structured nanocomposite used as a soil conditioner is beneficial to crops and soil. However, the relationship between the soil conditioner and soil microbes is unclear. We evaluated the soil conditioner’s impact on nutrient loss, pepper growth, soil improvement, and, especially, microbial community structure. High-throughput sequencing was applied to study the microbial communities. The microbial community structures of the soil conditioner treatment and the CK were significantly different, including in diversity and richness. The predominant bacterial phyla were Pseudomonadota, Actinomycetota, and Bacteroidota. Acidobacteriota and Chloroflexi were found in significantly higher numbers in the soil conditioner treatment. Ascomycota was the dominant fungal phylum. The Mortierellomycota phylum was found in significantly lower numbers in the CK. The bacteria and fungi at the genus level were positively correlated with the available K, available N, and pH, but were negatively correlated with the available P. Our results showed that the loss of nutrients controlled by the soil conditioner increased available N, which improved soil properties. Therefore, the microorganisms in the improved soil were changed. This study provides a correlation between improvements in microorganisms and the network-structured soil conditioner, which can promote plant growth and soil improvement.

## 1. Introduction

Crop production is expanding at a remarkable rate because of increased demand for food, which is of particular necessity for the world’s increasing population. Therefore, there is a need to produce more food from ever-limited land resources prepared by mixing attapulgite and PA. Soil is closely linked to agriculture-based economies, food security, and human health [1]. Plant growth and yield are limited by various factors, including soil, which is one of the most important parameters. Alterations to soil properties can result in enhanced plant production. Chemical fertilizer, which is widely used, can provide macro- and micronutrients in soil. Therefore, the application of fertilizer could promote the fast growth of crops and increase the yield of crops [2,3]. However, the amount of fertilizer used is constantly growing, and it is used in excess of what is needed. The use of extensive amounts of fertilizer has caused various negative effects on the environment; for example, excessive fertilizer use is a cause of greenhouse gas, water pollution, and soil degradation [4,5]. Synthetic nitrogen fertilizers, such as urea, contribute significantly to N_2_O emissions in the atmosphere [6]. Only 50–60% of nitrogen fertilizers are used by crops [7], and a portion of the rest runs off into water bodies [8]. Hence, it is important to supply nutrients to plants without damaging the environment [9]. In particular, there is an urgent need to develop fertilizers using innovative technology that can control the loss of nitrogen, increase the utilization efficiency of nitrogen, and decrease the level of environmental pollution [10].

Soil conditioners (SCs) are also called soil amendments and are an optional technology to increase crop production. An SC consists of materials that are added to soil that improve the soil conditions, especially by increasing nutrient availability [11], altering the soil physical structure [12], and stimulating microbial communities, which may enhance plant growth [13]. However, if an SC is added in excessive quantities, it may harm plant health, harm the environment, or enter the food chain [14]. Some materials used for controlled release are expensive [15], and for some materials, such as inhibitors of nitrification, actual field data are lacking [16]. Therefore, it is necessary for this new technology to not only increase the crop yield but also improve soil quality, and in particular, improve microbial communities.

In our previous study, a network-structured nanocomposite was fabricated that not only reduced the loss and effectively enhanced the utilization efficiency of nitrogen, but also modified the bacterial structure [17]. This nanocomposite, which could also be used as an SC, showed high biosafety and has the potential to reduce agricultural pollution. However, how SC causes changes in microbial communities and soil properties is unclear and needs to be studied further.

Soil microbes, including bacteria and fungi, are very important factors in an ecosystem. They are associated with many aspects of soil quality and health. Microbial communities provide different functions in soil, such as the cycling of nutrients, the degradation of various compounds, the promotion of plant growth, and the control of diseases [18,19]. A wide range of microbes have beneficial effects on soil. For example, arbuscular mycorrhizal (AM) fungi and plant-growth-promoting rhizobacteria (PGPR), including the nitrogen-fixing bacteria *Rhizobium* spp., belong to this group of beneficial microbes [19,20]. Microorganisms can mineralize soil nutrients to make them available for plants, bind soil particles together by secreting sticky polysaccharides, and coregulate the plant hormonal balance to help plants adapt to abiotic and biotic stressors [21,22]. Therefore, microbial diversity can be seen as a trait of soil quality that reflects nutrient efficiency and availability [23]. Soil microbial biomass, activity, and diversity are used as indicators of soil and ecosystem health [24].

High-throughput sequencing technology can provide deeper insight into the analysis of microbial communities [25,26]. It is an advanced method to research microbial communities because it can detect almost all species without culturing, even low-copy-number microorganisms [27]. Therefore, an in-depth analysis can be conducted from the sequencing results. This technique was used in this study to investigate microbial community changes in soil after applying SC.

Today, many studies have been reported regarding microbial community composition in soil [28]. However, the reason for the changes in microbial communities in soil treated by network-structured SC is not clear. Here, we addressed three aims: (1) evaluate the runoff of nutrients and plant growth caused by the nanonetwork-structured SC; (2) assess the effect of SC on the relationship between soil properties and microbial community structure; and (3) analyze the correlation between the network-structured SC and microorganisms. High-throughput sequencing technology was used to investigate the changes in microbial communities. The data could help understand the relationship between SC and microbial communities. We compared the changes in microbial communities at different taxonomic levels between a treatment in which the SC was administered and a treatment in which the soil was not amended. This study will help us understand the beneficial effects of our novel SC, which is a network-structured nanocomposite, on soil properties, microbial communities, and the growth of plants in greenhouses.

## 2. Materials and Methods

### 2.1. Preparation of Soil Conditioner and Leaching Assay

We purchased attapulgite powder (100–200 mesh) from Mingmei Co, Ltd. (Mingguang, China). Polyacrylamide (W. M., 5 million), urea (99%), and other chemicals were purchased from Shanghai Macklin Biochemical Co., Ltd. (Shanghai, China). Soil (20–50 mesh, pH 7.7, bulk density 1.5 g/cm^3^, available N 73.5 mg/kg, available P 87.8 mg/kg, available K 407.3 mg/kg) and sand (20–50 mesh, bulk density 1.5 g/cm^3^) were sourced from Donghua University (Shanghai, China). Deionized water was used in the experiments. Tap water was used in the greenhouse experiment.

The preparation of SC and the leaching assay were carried out as previously described [17] with minor modifications. In our previous study, the network-structured nanocomposite consisted of attapulgite, sodium humate, and polyacrylamide. Here, we modified the components. Sodium humate was removed. Briefly, SC with a weight ratio (W_attapulgite_:W_polyacrylamide_ = 50:1) was prepared by mixing attapulgite and PAM powders. Attapulgite and polyacrylamide in a weight ratio of 50:1 were mixed thoroughly to obtain the SC. A 50 g sand–soil mixture (Wsoil:Wsand = 3:7) was added to a 50 mL centrifuge tube with a 4 mm diameter hole at the bottom. A series of mixtures composed of urea (0.2 g) and SC (0.2 g, 0.4 g, 0.8 g) were prepared. Finally, the sand–soil mixture (10 g) covered the top of the SC–urea. A total of 50 mL of deionized water was added to the top of the system, and the urea concentration in the leachate was measured. The experiments were performed in triplicate.

### 2.2. Interaction Analyses of Soil Conditioner

A scanning electron microscope (SEM; SU8220, Hitachi High-Technologies Co., Tokyo, Japan) was used to study the morphology of the SC. A Fourier transform infrared spectrometer (FTIR, iS10, Nicolet Co., La Crosse, WI, USA) and a TTRIII X-ray diffractometer (XRD; TTRIII, Rigaku Co., Tokyo, Japan) were used to analyze the structure and chemical state of the SC. A thermogravimetric analyzer (DSCQ2000, TA Co., Valencia, CA, USA) was used for the thermal gravimetric analysis (TGA) and differential thermal analysis (DTA) of the SC. The Brunauer–Emmett–Teller (BET) specific surface areas of the attapulgite and SC were analyzed using an automatic surface area and a pore analyzer (BELSORP-max, MicrotracBEL, Osaka, Japan). The pore structure of the SC was measured based on the Barrett–Joyner–Halenda (BJH) method.

### 2.3. Greenhouse Experiment and Design

The greenhouse experiment took place in Qianbu County (36°55’ N, 118°40’ E), Shouguang City, Shandong Province, China. This site belongs to the Shouguang vegetable industry holding group, which is a well-known vegetable industry operator in China. To obtain a higher yield and quality of vegetables, many plastic greenhouses were constructed here over 10 years to grow cucumbers, pepper, tomatoes, and other vegetables. The average annual temperature of the site is approximately 12.7 °C, the average annual amount of precipitation is approximately 593.8 mm, and the total amount of sunshine is 2548.8 h. The temperature of the studied greenhouse was kept at 28 °C to 30 °C, and the relative soil moisture was kept at approximately 50%. The soil’s main properties were as follows: pH 7.84, available N 73.5 mg/kg, available P 87.8 mg/kg, available K 407.3 mg/kg, and organic matter 26.1 g/kg. The previous vegetable grown in our greenhouse was tomato. To investigate the soil and economic benefits that SC had on vegetable products, square pepper (*Capsicum annuum* L.) WeiLi was grown in the greenhouse in 2021.

Plots of 8.25 × 6.00 m^2^ were used to plant the square peppers. There were three experimental treatments, and one control treatment was used based on the different amounts of SC used. Three replicates were designed for each treatment. A completely randomized experimental design was implemented. The SC needed to be mixed with base fertilizer evenly for use. The conditions were as follows: only fertilizer without soil conditioner, used as the control treatment (CK); 100 kg ha^−1^ soil conditioner (SC1); 200 kg ha^−1^ soil conditioner (SC2); and 400 kg ha^−1^ soil conditioner (SC3). The base fertilizer was composed of urea (150 kg ha^−1^), ammonium phosphate (450 kg ha^−1^), and potassium sulfate (300 kg ha^−1^). All of these fertilizers were sprayed on the surface and tilled into the soil in a layer approximately 20 cm below the topsoil.

### 2.4. Plant Growth, Soil Collection, and Properties

A total of 165 square pepper seedlings were transplanted into each plot. The plant distance and row distance were 50 and 60 cm, respectively. The management was the same for all treatments during the whole planting period. The experiment was performed in replicates, with 3 plots per treatment. The plant height and leaf chlorophyll content (measured using chlorophyll meter SPAD 502 Plus) were measured before the flowering stage. The fruit length and weight were measured at the end of the maturity stage. The number of earthworms in the soil was measured after the fruit stage.

Soil samples according to the “S” shape were collected after the fruit stage [29]. For the surface soil (0–20 cm) in each plot, five bulk plant soils (20 cm in depth, 5 cm in diameter) were collected. Then, the five bulk soils were pooled together to form a composite sample. There were three replicates for each treatment, so we obtained three composite samples. The soil samples were stored at 4 °C and used to analyze the soils’ physiochemical properties. The following soil properties were measured: pH, total organic matter (OM), total salt (TS), available nitrogen (AN), available phosphorus (AP), available potassium (AK), soil capacity, porosity, and water retention. The soil properties were determined using previously reported methods [30].

The rhizosphere samples were collected with the soil samples in the meantime, according to the “S” shape in the previous method, with minor modifications [31,32]. The plant closest to each random point was selected. Therefore, each rhizospheric soil sample was obtained from 5 plants. The bulk soil was removed from the roots of the plants, and the soil that closely adhered to the surface of the root was collected. The soil samples were used for the microbial analysis. Three replicates were used for each treatment.

Student’s *t*-test was carried out to analyze statistical significance with GraphPad Prism software (Version 9.3.0).

### 2.5. High-Throughput Sequencing of the Microbe Communities and Data Processing

Total soil DNA was extracted using a TIANamp Soil DNA Kit (TIANGEN biotech (Beijing) Co., Ltd., Beijing, China). The purity and concentration were quantified at OD260/280 nm using a NanoDrop spectrophotometer (Thermo Scientific, Waltham, MA, USA), and DNA was kept at −80 °C for further studies.

The purified DNA was amplified in the V3-V4 region of the bacterial 16S rRNA gene using the 341F (5′-CCTAYGGGRBGCASCAG-3′)/806R (5′-GGACTACHVGGGTWTCTAAT-3′) [33] primers and the ITS1-5F region of the fungal rDNA gene with ITS5-1737F (5′-GGAAGTAAAAGTCGTAACAAGG-3′) and ITS2-2034R (5′-GCTGCGTTCTTCATCGATGC-3′) [34]. The PCR product was separated via 2% agarose gel electrophoresis and purified using a Qiagen Gel Extraction Kit (Qiagen, Germany). Library construction and sequencing were conducted by Novogene Co., Ltd. (Novogene Co., Ltd., Beijing, China, https://www.novogene.com/, accessed on 1 August 2022). The sequencing library was built using a TruSeq DNA PCR-Free Sample Preparation Kit (Illumina, San Diego, CA, USA). The library quality was assessed on the Qubit@ 2.0 Fluorometer (Thermo Scientific) and Agilent Bioanalyzer 2100 system. Then, the Illumina NovaSeq platform was used to sequence the library.

Paired-end reads were merged using FLASH (V1.2.7, http://ccb.jhu.edu/software/FLASH/, accessed on 1 August 2022) [35], and the splicing sequences were called raw reads.

The raw read data were submitted to Science Data Bank (https://doi.org/10.57760/sciencedb.07079, accessed on 1 August 2022. DOI:10.57760/sciencedb.07079).

To obtain high-quality clean reads, QIIME (V1.9.1, http://qiime.org/scripts/split_libraries_fastq.html, accessed on 1 August 2022) was used to filter the raw reads following previous reports [36].

To obtain effective reads, the UCHIME algorithm (http://www.drive5.com/usearch/manual/uchime_algo.html, accessed on 1 August 2022) was used to detect and remove chimera sequences [37].

### 2.6. Statistical Analysis

To cluster the clean reads into operational taxonomic units (OTUs) with ≥97% similarity, Uparse software (Uparse v7.0.1001, http://drive5.com/uparse/, accessed on 1 August 2022) was used. To annotate taxonomic information, bacterial OTUs were aligned against the SILVA SSUrRNA database (https://www.arb-silva.de/, accessed on 1 August 2022) [38]. The fungal OTUs were aligned against the UNITE database (https://unite.ut.ee/, accessed on 1 August 2022) [39] using QIIME software (Version 1.9.1, http://www.drive5.com/usearch/manual/uchime_algo.html, accessed on 1 August 2022) [40].

To determine the phylogenetic relationships of different OTUs, MUSCLE software (Version 3.8.31, http://www.drive5.com/muscle/, accessed on 1 August 2022) was used [41]. The OTUs were standardized, and the subsequent diversity analysis was performed on the basis of the output normalized data.

To understand the changes in microbial structure, alpha diversity and beta diversity were analyzed. Alpha diversity was calculated by QIIME (Version 1.9.1). The results were displayed and differences were analyzed using R software (Version 2.15.3). A heatmap was drawn from the OTUs’ relative abundance using the Vegan Package in R 2.4 (https://cran.r-project.org/web/packages/vegan/, accessed on 1 August 2022). QIIME software (Version 1.9.1) was used for the beta diversity analysis to calculate unweighted UniFrac distances, and R software (Version 2.15.3, R ade4 package and ggplot2 package) was used to draw graphs. The unweighted pair group method with arithmetic mean (UPGMA) was used for hierarchical clustering analysis. To reveal the variation in the microbial communities of the different samples, principal coordinate analysis (PCoA) was performed based on the unweighted UniFrac distances. Linear discriminant analysis (LDA) effect size (LEfSe) (http://huttenhower.sph.harvard.edu/lefse/, accessed on 13 March 2023) was used to detect the potential biomarkers by utilizing LEfSe software (LDA Score = 4). Student’s *t*-test was used to calculate the differences between the different microbial abundance data. The statistical significance was set at *p* < 0.05.

Spearman correlation (*p* < 0.05) and redundancy analysis (RDA) were used to explore the relationship between environmental factors and species richness and microbial community structure.

PICRUSt and FunGuild analyses were conducted to predict the functions of the microbial community. PICRUSt is a bioinformatics package that uses marker genes to predict the functional content of microorganisms [42]. The potential functions of each sample were predicted based on 16S rRNA sequencing data. A heat map of the Kyoto Encyclopedia of Genes and Genomes (KEGG) level 3 functional pathways was created using the R package (Version 2.15.3). FUNGuild is a tool that can be used to obtain the ecological functions of fungi based on 18S or ITS rRNA sequencing data [43,44].

## 3. Results

### 3.1. Effect of Controlling Urea Leaching Using the Soil Conditioner

As shown in Figure 1A, the effect of different amounts of SC on the leaching loss of urea in sand–soil mixtures was investigated. Figure 1B indicates that the concentration of urea in the leachate obviously decreased with increasing amounts of SC. An amount of 0.8 g SC was shown to have the best ability to control the loss of urea. This result illustrated that the SC possessed a strong ability to control the loss of urea and increased the utilization efficiency of urea.

### 3.2. Interaction Characteristics of Soil Conditioner Components

To elucidate the mechanism of the SC that enabled it to control loss ability, the microstructure of the SC system was investigated. As shown in Figure 2A,B, attapulgite nanorods aggregated to form several bunches due to the high surface activity and nanoscale effect. After it was loaded with polyacrylamide, the SC (Figure 2C,D) became a micro/nanonetwork due to bridging and netting effects, which were probably caused by the hydrogen bond between the attapulgite and polyacrylamide. Thus, it was shown that the SC possessed a high capacity to decrease the loss of urea because of the SC micro/nanonetwork.

XRD measurements were used to analyze the interactions in the SC system (Figure 3A). The characteristic peaks in the spectrum of the SC indicated that the attapulgite and polyacrylamide combined successfully. Additionally, no new characteristic peak was generated after the combination of attapulgite and polyacrylamide, indicating that intercalation did not exist between the attapulgite and polyacrylamide.

The interactions in the SC/urea system were analyzed by FTIR measurement (Figure 3B). Compared with the characteristic peaks of attapulgite and polyacrylamide, the peaks of the SC redshifted, and the intensity of the peaks decreased (Figure 3A). After the combination of SC and urea in the SC/urea system, the peaks of urea shifted, and the peak intensity decreased [17,45]. Meanwhile, the peak intensity of the SC clearly decreased in the SC/urea system. These results illustrated that hydrogen bonds probably linked the SC (-OH) and urea (N-H).

TG-DTA analysis was used to study the thermal stability of the attapulgite, polyacrylamide, and SC. As shown in Figure 3C, the initial weight loss region (51–183 °C) probably corresponded to water. The remaining weight loss regions (183–325 °C, 325–571 °C, and 571–800 °C) were probably due to organic matter degradation in the attapulgite. Figure 3D indicates that the initial weight loss (51–328 °C) may be due to water loss. The latter weight loss region (328–800 °C) was ascribed to the probable decomposition of polyacrylamide. As shown in Figure 3C–E, the weight loss regions of the SC were consistent with those of attapulgite because the SC contained 98% attapulgite. The weight loss values of attapulgite, polyacrylamide, and SC were 19.1% (Figure 3C), 92.7% (Figure 3D), and 19.5% (Figure 3E), respectively. These values illustrate that the SC included approximately 98% attapulgite, and thus, the results of the TG-DTA analysis were consistent with the weight ratio of attapulgite:polyacrylamide in the SC. As shown in Figure 3F, many pores appeared, with a size distribution of 5–60 nm in the SC (inset of Figure 3F, BJH method analysis) and a high BET specific surface area of 212.37 m^2^/g, which was beneficial for the adsorption of urea.

These results showed that the network structure of the SC linked the urea by hydrogen bonds.

### 3.3. Effects of Soil Conditioner on Plant Growth

A greenhouse test of the SC was performed in Shouguang City, Shandong Province, to illustrate the effects of the SC on pepper and soil. As shown in Appendix A, the SC had a significant positive influence on the growth of pepper. Figure 4A–D indicates that with the increase in the amount of SC, the plant height, SPAD, and fruit length and weight increased. These results suggested that SC can increase the yield of pepper.

### 3.4. Effect of Soil Conditioner on Soil Characteristics

Based on the characteristic analysis of the SC and plant growth, we selected SC3 to determine the characteristics of the soil. Table 1 shows that the SC3 soil had a high capacity to increase organic matter, available nitrogen, and potassium. This was probably because the SC possessed a nanonetwork structure that decreased the loss of nutrients and the degradation of organic matter. The SC reduced the soil capacity (4.55%) and increased the soil porosity (12.57%), indicating that the soil became loose after treatment with SC3 (Table 1). In addition, the water retention rate was enhanced by 10.04% compared to that in the treatment without SC, probably because of the nanonetwork structure of the SC (Table 1).

Earthworms can be used as indicators to assess soil quality [46,47]. The abundance of earthworms at a particular site can indicate high soil quality. We collected and recorded the number of earthworms in the different treatment soils. The number of earthworms increased gradually with the increase in the amount of SC, indicating that SC promoted the prevalence of earthworms (Appendix A). After treatment with SC, there was a significant number of earthworms in the SC3 soil (Appendix A).

These results illustrated that SC can be used as a promising soil conditioner in soil.

### 3.5. The Microbe Communities Identified Using High-Throughput Sequencing

Based on the results presented above, we selected SC3 to analyze the microbial communities via high-throughput sequencing.

#### 3.5.1. Alpha Diversity of the Microbial Community

After a quality control process, a total of 510,150 clean reads from the V3-V4 region of the 16S rRNA were obtained. The effectiveness of these reads was more than 91.08%. A total of 480,918 clean reads were obtained from the ITS1-5F region. The effectiveness of these reads was more than 96.07% (Appendix A). The subsequent analysis was performed on the basis of the normalized data.

The Venn diagram (Figure 5) shows the differences between CK and SC3 at the OTU level. In the bacteria (Figure 5A), there were 1386 common OTUs in CK and SC3, but there were 289 unique OTUs in CK and 420 unique OTUs in SC3. In the fungi (Figure 5B), there were 517 common OTUs in CK and SC3, but there were 155 unique OTUs in CK and 178 unique OTUs in SC3. The results indicate that both the bacterial and fungal diversity in SC3 were richer than those in CK. The bacterial diversity changed more than the fungal diversity. This implies that the SC improved the soil environment to accumulate more bacterial and fungal diversity.

The rarefaction curves tended to be flat (Appendix A), suggesting that a reasonable number of individual samples were taken. The high Good’s coverage values indicate that the sequencing depth was adequate for the community analysis (Table 2). Therefore, the data were sufficient for the analysis of microbial communities. Simultaneously, the rarefaction curves revealed that the microbial community richness in the SC3 samples was higher overall than that in the CK samples.

The alpha diversity indices indicated that there were significant differences between the CK and SC3 treatments (Table 2). Compared to the CK, the richness indices (including oberved_species, Chao1, and ACE) and diversity indices (including Shannon, Simpson, and PD_whole_tree) were higher in the treatment.

For bacteria, the Chao1, ACE, Shannon, and Simpson indices were significantly higher in SC3 than in CK. The oberved_species and PD_whole_tree were also higher in SC3 but were not significantly different from the CK. There was little difference in the fungi. Except for the Simpson index, which displayed no significant difference, the Chao1, ACE, Shannon, and PD_whole_tree were significantly higher in SC3 than in CK. From these results, the richness and diversity were significantly higher in SC3 than in CK in terms of both bacteria and fungi. These results indicated that the diversity of the microbial community was affected by SC3.

#### 3.5.2. Comparison of Bacterial Community Composition Based on 16S rRNA Sequencing

All bacterial sequences were classified at both the phylum and the genus level (Figure 6). The top 10 relative abundances of the bacteria in each sample at the phylum level were analyzed using the UPGMA clustering method (Figure 6A). The relative average abundance of the top 10 bacteria at the phylum level in the CK and SC3 treatments were Pseudomonadota (30.65 and 20.31%), Actinomycetota (23.75% and 17.23%), Bacteroidota (12.91% and 5.11%), Acidobacteriota (1.33% and 9.81%), Chloroflexi (2.35% and 4.62%), Gemmatimonadota (4.73% and 1.69%), Bacillota (2.36% and 3.15%), Myxococcota (2.25% and 2.54%), and Verrucomicrobiota (1.17% and 1.25%). The relative average abundance of unclassified_Bacteria at the phylum level accounted for 14.02% in CK and 23.76% in SC3. The rest of the phyla were presented in “other phyla”, including Planctomycetota, Nitrospirota, and Cyanobacteria. Additionally, the average proportion of “other phyla” accounted for 4.49% in CK and 10.54% in SC3. The bacteria with high relative abundance, including Pseudomonadota, Actinomycetota, and Bacteroidota, were lower in SC3 than in CK. The abundance levels of Pseudomonadota, Actinomycetota, and Bacteroidota were significantly higher in SC3 than in CK (*p* < 0.05). The UPGMA analysis results also showed similar samples in each treatment.

More details regarding the relative abundance of the bacterial community at the genus level are shown in a heatmap (Figure 6B). The top 35 bacteria at the genus level in the CK and SC3 treatments are presented. In SC3, *Sphingomonas*, *Blastococcus*, *RB41*, etc., were the dominant genera. In CK, the most representative genera were *Paeniglutamicibacter*, *Luteimonas*, *Pedobacter*, etc. The relative abundance levels of *Massilia*, *Sphingomonas*, *Aeromicrobium*, etc., were higher in SC3 than in CK. Meanwhile, the levels of unclassified *Gemmatimonadacae*, *Luteimonas*, *Galbitalea*, etc., were higher in CK than in SC3. Some genera had similar average relative abundance levels between CK and SC3, such as Flavisolibacter (0.72% and 0.62%) and Nocardioides (0.84% and 0.85%). This suggests that the changes in relative abundance occurred after the SC3 treatment. These changes may have occurred because the SC modified the soil ecosystem.

Principal coordinate analysis (PCoA) was performed to reveal the variation in the bacterial communities among the different samples (Appendix A). The PCoA significantly separated the CK samples from the SC3 samples, indicating that the two treatments had their own specific characteristic microbial community. The heat map of beta diversity also indicated that the microbial communities of the samples were clearly different between CK and SC3 (Appendix A). Both the PCoA and heatmap results showed samples that were similar in their bacterial compositions in each treatment. The bacterial compositions of the samples in the CK were different from those in SC3. The results were also the same as the results of the UPGMA analysis (Figure 6A).

To examine the taxonomic bacteria with significant abundance differences between CK and SC3, linear discriminant analysis effect size (LEfSe) was used for biomarker analysis. The results showed that there were 35 distinct bacterial taxa with LDA values greater than 4.0 (Figure 7A). In SC3, three phyla, five classes, four orders, six families, seven genera, and eight species were enriched. Meanwhile, one genus (*Luteimonas*) and one species (*Luteimonas* mephitis) were enriched in the CK. LEfSe analysis indicated that SC3 influenced the bacterial composition of the soil.

#### 3.5.3. Comparison of Fungal Community Composition Based on ITS Sequencing

The relative abundance of the fungi at the phylum and genus levels is shown in Figure 8. The top 10 relative abundance levels of the fungal community in each sample at the phylum level were analyzed using the UPGMA clustering method (Figure 8A). The Ascomycota phylum was the most abundant in the two treatments (50.54% and 51.83%). The abundance of the Mortierellomycota phylum was significantly lower in CK than in SC3 (3.14% and 17.4%). Additionally, the abundance of the Basidiomycota phylum was significantly higher in the CK (14.17% and 9.42%). The average proportion of “other phyla” accounted for 30.56% in CK and 19.00% in SC3. The UPGMA analysis results also showed similar samples in each treatment.

The relative abundance levels of the top 35 fungi at the genus level in CK and SC3 are presented as a heat map (Figure 8B). In contrast to SC3, *Pseudogymnoascus*, *Thelebolus*, *Botryotrichum*, etc., were the dominant genera in the CK. In the SC3, *Mortierella*, *Thysanorea*, *Calyptella*, etc., were the dominant genera. The abundance of some fungi in SC3 was higher than that in CK, including *Alternaria*, *Acremonium*, *Mortierella*, etc. The unclassified_*Eurotiales*_sp. was only found in the SC3. This suggested that SC3 changed the relative abundance of fungi.

The variation in and relationship between fungal communities were analyzed using PCoA (Appendix A). There was 68.72% variance in the first principal coordinate axis (PCo1) and 18.19% variance in the second principal coordinate axis (PCo2). The relationships between different samples were also the same as those shown in the UPGMA analysis (Figure 8A). The heat map of beta diversity also indicated that the fungal communities of the samples were obviously different between CK and SC3 (Appendix A).

LEfSe was conducted to identify the taxa with significant differences between CK and SC3 (Figure 9). The results showed that there were 46 fungal taxa with distinguishable differences with LDA values greater than 4.0 (Figure 9A). From the LEfSe analysis, one phylum, two classes, five orders, five families, seven genera, and seven species were enriched in SC3, while three classes, four orders, four families, four genera, and four species were enriched in CK. These results indicated that fungal compositions in soil are influenced by the use of SC3.

#### 3.5.4. Correlation Analysis between Microbes and Environmental Parameters

Redundancy analysis (RDA) was conducted to identify the relationships between microbial compositions at the genus level and environmental factors (Figure 10). Four parameters—pH, available phosphorus (AP), available potassium (AK), and available nitrogen (AN)—were selected for RDA. Organic matter content was not included in our analysis because available nutrient loss could be controlled by SC3. We wanted to find the relationships between soil properties altered by SC3′s control of nutrient loss and the changes in microbial communities.

The physicochemical properties of the soils are shown in Table 1. The AN and AK contents were higher in SC3 than in CK, while the AP content was lower in SC3 than in CK. SC3 had little effect on the soil pH value.

The compositions of bacterial genera and fungal genera in the SC3 samples were both positively correlated with AK, AN, and pH. Meanwhile, the microbial community structures in CK samples, both bacterial genera and fungal genera, were positively correlated with AP (Figure 10). Changes in microbial communities relied on the AN, AK, and AP. The results showed that microbial communities are positively correlated with AN and AK, and they may be affected by SC.

Spearman correlation analysis was used to study the correlations between environmental variables and the microbial abundance in genera (Figure 11). In terms of bacterial abundance (Figure 11A), *Massilia* and *Blautia* were positively correlated with AN, while *Pontibacter* and *Paeniglutamicibacter* were negatively correlated. *Truepera*, *Flaviaesturariibacter*, *Gemmatimonas*, *Arthrobacter*, and *Paeniglutamicibacter* were positively correlated with AP, while *Massilia* and *Blautia* had negative correlations. *Kribbella*, *Aeromicrobium*, *Massilia*, and *Blautia* were positively correlated with AK, but *Truepera*, *UTBCD1*, and *Paeniglutamicibacter* were negatively correlated with AK. In terms of fungal abundance (Figure 11B), *Myrothecium*, *Acremonium*, and *Mortierella* were positively correlated with AN, but unclassified_*Leotiomycetes*_sp., *Trichosporon*, *Leucosporidium*, and *Cutaneotrichosporon* were negatively correlated. Unclassified_*Leotiomycetes*_sp., *Leucosporidium*, *Solicoccozyma*, and *Thelebolus* were positively correlated with AP, while *Titaea* had a negative correlation. *Myrothecium*, *Acremonium*, and *Titaea* were positively correlated with AK, but unclassified_*Leotiomycetes*_sp. and *Leucosporidium* had negative correlations with AK.

These results showed a correlation between AN, AK, AP, and the microbial community. The positive correlation between AN, AK, and the microbial community may be caused by the SC.

#### 3.5.5. Predicted Functional Gene Analysis for Microbial Communities

PICRUSt [42] was used to predict functional analyses for bacterial communities of all samples. A heat map of the 35 KEGG (level-2) pathways with relatively high abundance levels is shown in Figure 12A. KEGG pathways including replication and repair, nucleotide metabolism, signaling molecules and interaction, and cancers were significantly higher in abundance in the CK, while energy metabolism, metabolism of terpenoids and polyketides, metabolism of cofactors and vitamins, and endocrine system were significantly higher in abundance in the SC3 (Figure 12A and Appendix A).

FUNGuild [43] was used to predict functional analyses for fungal communities in all of the samples. The relative abundance levels of 26 fungal functional guilds (including others), such as Animal_Pathogen-Soil_Saprotroph, Endophyte-Plant_Pathogen-Wood_Saprotroph, and Plant_Pathogen-Soil_Saprotroph-Wood_Saprotroph, were detected (Figure 12B). The “Animal_Pathogen-Soil_Saprotroph” guild had a significantly higher abundance level in the CK. However, the “Endophyte-Plant_Pathogen-Wood_Saprotroph” and ”Plant_Pathogen-Soil_Saprotroph-Wood_Saprotroph” guilds had relatively high abundance levels that increased significantly in SC3 (Figure 12B and Appendix A). The predicted functional gene analyses for microbial communities from the KEGG (level-2) pathways and guilds were significantly different in CK compared with SC3.

## 4. Discussion

Fertilizers represent an indispensable factor in current agriculture to sustain soil fertility and crop productivity. However, the nutrient use efficiency of fertilizers is constantly low [48]. Fertilizers left in soil due to overuse and runoff can cause a variety of environmental problems, including the pollution of water bodies and changes in soil quality [49]. The application of nanotechnology can promote the nutrient use efficiency of crops and decrease the negative effects of synthetic fertilizers in agriculture. Due to their wide-ranging environmental applications, a number of nanomaterials have been used in agriculture [50,51,52]. As has been reported, the application of SC is a strategy that can be used to increase crop yields and improve the quality of soil [11,53,54]. In a previous study, it was shown that a fertilizer synergist could effectively inhibit hydrolysis, reduce loss, and enhance utilization efficiency [17]. In this study, we generated a nanonetwork-structured SC composed of attapulgite and polyacrylamide and investigated the effect of the SC on pepper agronomic traits, soil properties, and the structure of microbial communities. Based on the results, we assessed and evaluated the effect on the change in microbial communities after nanomaterials were applied to control the loss of nitrogen. To our knowledge, this is the first report regarding the ability of this nanonetwork-structured SC, composed of attapulgite, to alter microbial communities. Our study provides an understanding of the mechanism that allows network-structured nanocomposites to alter microbial structures.

### 4.1. Material Characteristics of Nanostructured Soil Conditioner and Characteristics of Reducing Nitrogen Loss

We determined the effect of different amounts of SC on the control of the loss of nutrients and observed the microstructure of the SC. The amount of SC influenced the control loss efficiency (Figure 1). In view of the high surface activity and nanoscale effect, attapulgite rods tended to form several bunches with abundant small pores among the rods. After the addition of polyacrylamide, a micro/nanonetwork was formed through bridging and netting effects, probably driven by the existence of hydrogen bonds between attapulgite and polyacrylamide (Figure 2 and Figure 3A,B). Urea could be loaded into the network to decrease the level of loss. This result is the same as our previous research regarding the microstructure of a fertilizer synergist. Our FTIR spectroscopy and XRD analyses indicated that the H bonds likely played a key role in the formation of the SC network. The thermogravimetric analysis (TGA) and differential thermal analysis (DTA) of the SC (Figure 3C–F) revealed that SC underwent a three-stage thermal decomposition process. The difference between the SC and fertilizer synergist is that sodium humate was mixed in the fertilizer synergist. Sodium humate is an efficient urease inhibitor and can be loaded into the small pores to form part of the fertilizer synergist. From previous research, it can be seen that the absence of sodium humate has no effect on the micro/nanonetwork structure of the SC [17]. These results indicated that the network structure linked by H bonds between polyacrylamide and attapulgite has a good ability to store nutrients and control nutrient loss.

### 4.2. Effect of SC on Soil Physicochemical Properties

Generally, soil conditioners are added to soil to improve its condition in terms of structure and nutrients with the goal of increasing plant growth. Therefore, we recorded the growth of pepper and the changes in soil properties. In our study, the growth of the treated peppers was advanced, and the yield was higher. The application of SC caused no change in the soil pH. The other soil properties, such as organic matter and available N and K contents, were increased through the application of SC. In particular, we compared the soil capacity, porosity, and water retention between the control and SC3 soil. These three physicochemical traits were also improved (Table 1). As the prevalence of earthworms is generally associated with good soil, we counted the number of earthworms. In the SC-improved soil, the abundance of earthworms was higher than that in the CK soil (Appendix A). As the use of nanotechnology has increased, nanomaterials have contributed to agriculture [55]. Attapulgite has many nanoscale channels, giving it unique physical and chemical properties. It has been widely used as an additive of nanocomposites [56], as the supporter of active nanomaterials, and to control nitrogen or pesticide loss [2,17,57]. In this study, the application of SC composed of attapulgite resulted in higher nutrient availability; better soil capacity, porosity, and water retention; and increased plant yield. These results were in agreement with previous reports on soil conditioners [58,59].

For better application of SC, further research should be conducted in field conditions. The results may be different between greenhouse and field due to the different environments (soil type, pH, location, water, fertilizer, and so on). The stability of SC in controlling nutrient loss should be researched in various soils.

### 4.3. Effect of SC on Microbial Community Structure

In this study, microbial abundance and diversity in soil, including bacteria and fungi, were assessed using high-throughput sequencing. We found that the richness and diversity of bacteria were significantly higher than those of fungi (Table 2). This implies that bacteria are dominant colonizers in this soil. The alpha diversity was also shown to differ significantly between the CK and SC treatments, both in terms of bacteria and fungi. Meanwhile, the UPGMA and PCoA results in this study indicated that there were different characteristic microbial communities between CK and SC3 (Figure 6A, Appendix A, Figure 8A and Appendix A).

#### 4.3.1. Effect of SC on Bacterial Microorganisms

Bacterial communities are the most diverse and abundant groups in soil. Pseudomonadota, Actinomycetota, Bacteroidota, Gemmatimonadota, and Chloroflexi were the dominant phyla in CK and SC3. The predominant phyla in our study are almost the same as those in previous studies. In marine ecosystems, Pseudomonadota, Bacteroidota, and Bacillota are the dominant phyla. In apple orchard soils, Pseudomonadota, Actinomycetota, and Acidobacteria are the main phyla. On seaweed surfaces, Pseudomonadota and Bacillota are the most abundant phyla. In SC3 soil, the relative abundance levels of Pseudomonadota, Actinomycetota, Bacteroidota, and Gemmatimonadota were reduced, but the abundance levels of Acidobacteriota, Chloroflexi, Bacillota, and Myxococcota were increased (Figure 6A). Pseudomonadota are responsible for the symbiotic nitrogen fixation of legumes and biodegradation of polycyclic aromatic hydrocarbons [60,61]. Actinomycetota could be responsible for the decomposition of all sorts of organic substances, the production of bioactive metabolites, and the biological control of soil environments [62]. Bacteroidota are well-known degraders of organic matter [63]. Gemmatimonadota seem to be frequently associated with plants and the rhizosphere and have the capacity for anoxygenic photosynthesis [64]. A previous study showed that soil amendment alters the abundance of phyla. The relative abundance level of Pseudomonadota was increased, but the levels of Acidobacteria, Actinobacteria, Gemmatimonadetes, and Chloroflexi were reduced [59]. This differs somewhat from our results. At the genus level, the relative abundance levels of some bacteria, such as *Massilia*, *Sphingomonas*, *Aeromicrobium*, *Kribbella*, and *Gaiellawere*, were higher in SC3. Additionally, the relative abundance levels of *Nitrosospira*, *Luteimonas*, *Galbitalea*, and *Gemmatimonas* were lower in SC3 than in CK (Figure 6B). *Massilia* is abundant in the rhizosphere and can interact with pathogenic fungi by degrading chitin [65,66]. *Sphingomonas* functions in plant tolerance to abiotic stress and the biodegradation of environmental contaminants [67,68]. *Aeromicrobium* has been reported to possess the ability to degrade organic matter [69]. *Kribbella* and *Gaiella* [70,71] may inhibit pathogens. In the present study, soil improvement reduced the relative abundance of bacterial genera, such as *Gemmatimonas* and *Nitrosospira* [72]. Our results are similar to these reports. We noted that in our study, *Luteimonas* was enriched in the CK (Figure 7). *Luteimonas* was reported as a denitrifying genus in a wastewater treatment system [73]. These results may have shown that the soil environments were improved by the SC.

#### 4.3.2. Effect of SC on Fungal Microorganisms

Soil fungi have many functions, including as biological controllers, ecosystem regulators, and participants in the conversion of organic matter [74]. In our results, Ascomycota, Basidiomycota, and Mortierellomycota were the dominant phyla in CK and SC3 (Figure 8A). These predominant phyla in our results are consistent with the results of a previous study [75]. The Ascomycota phylum was the most dominant in all of the agricultural soil and plays an ecological role as a decomposer [76]. Basidiomycota can degrade different components of wood, and they are part of a key process in carbon recycling [77]. The abundance of the Mortierellomycota phylum in SC3 was significantly higher than that in CK. Mortierellomycota can dissolve mineral phosphorus and secrete oxalic acid to increase soil nutrient contents [78,79]. In our results, the AP content was lower in SC3 than in CK. There was a negative correlation between Mortierellomycota and AP. We speculate that the mineral phosphorus in soil was dissolved by Mortierellomycota, and then, the available P was absorbed by plants to improve their growth. This needs to be researched in future work. At the genus level, *Mortierella*, *Thysanorea*, *Calyptella*, *Pseudogymnoascus*, etc., were the dominant genera in the SC. The relative abundance levels of *Alternaria*, *Acremonium*, *Mortierella*, *Paraphoma*, and *Trichoderma* were higher in SC3 than in CK (Figure 8B). *Thysanorea* belongs to the Ascomycota phylum and may play a key role in decomposition in soils and increasing nitrogen [80]. *Calyptella* belongs to the Basidiomycota phylum and is confirmed to be an ectomycorrhizal mutualist, which protects plants [81]. *Mortierella* was shown to be the dominant fungus in fertilizer-treated soils and was affected by the nutrients in the soil. *Pseudogymnoascus* decays matter in cool environments. *Trichoderma* and *Acremonium* levels can increase in fertilized soils, and they can be used in biocontrol. In addition to beneficial fungi, the levels of pathogenic fungi such as *Fusarium*, *Alternaria*, *Botryotrichum*, and *Pseudogymnoascus* increased in SC3 (Figure 8B). We speculated that there was an increase in fungal diversity in the SC-improved soil, including beneficial and pathogenic fungi. However, the richness and diversity levels of beneficial fungi were higher than those of pathogenic fungi in the SC3 soil. Therefore, SC improves soil.

### 4.4. Relationship between Soil Physicochemical Properties and Microbial Community after Application of SC

It was reported that microbial communities in soil strongly correlate to soil physicochemical properties [82]. In our study, the soil physicochemical properties were improved by the addition of SC (Table 1). The organic matter content was absent from our analysis because this study mainly focused on the control of the loss of available nutrients. The pH of soil significantly influences the structure of microbial communities. However, our SC did not affect the soil pH. The soil properties showed a correlation with the microbial community (Figure 10 and Figure 11). The compositions of the bacterial genera and fungal genera in the SC3 samples were both positively correlated with AK, AN, and AP (Figure 11). At the genus level, bacteria such as *Paeniglutamicibacter* (Actinomycetota phylum), *Blautia* (Firmicutes phylum), and *Massilia* (Pseudomonadota phylum) were significantly correlated with AK, AN, and AP; fungi such as *Leucosporidium* (Basidiomycota phylum), *Acremonium* (Ascomycota phylum), and *Myrothecium* (Ascomycota phylum) were significantly correlated with AK and AN. Unclassified_*Leotiomycetes*_sp. (Ascomycota phylum), *Leucosporidium* (Basidiomycota phylum), *Solicoccozyma* (Basidiomycota phylum), and *Thelebolus* (Ascomycota phylum) were positively correlated with AP. We proved that SC could control the loss of nutrients (Figure 1). We speculated that the AP and AK levels were also controlled by the nanonetwork structure of the SC. The effect of soil carbon on the microbial community was not determined. In fact, microorganisms play an important role in the soil carbon cycle. Microbial biomass can increase soil carbon storage. Meanwhile, microorganisms are commonly associated with soil organic carbon degradation [83,84]. Previous reports have found that microbial interactions with soil C and N could be affected by many factors. The factors include litter input, fine roots, soil organic carbon, inorganic N, soil moisture, temperature, and pH [85,86]. SC can control the loss of nutrients that increase available N in soil. The increased N may induce P limitation and potentially affect C cycling. We speculated that the C:N ratio was changed by the SC in soil. This may be the cause of the changes in the microbial community. This needs more study in the future.

In our study, SC affected soil fertility by controlling nutrient loss, with the potential to change the soil microbial quantity. The altered trends in the microbial community in different fields should be studied in further work.

### 4.5. Predicted Functional Analysis for Microbial Communities

Compared with the CK, the soil properties and the bacterial community structure were altered by the administration of SC. Considering that the SC treatment’s performance may have been affected by functional microorganisms, we analyzed the functional genes of the microbial communities. Bacterial results were obtained from the PICRUSt analysis. The number of genes related to cell growth and death, endocrine system, and energy metabolism in the SC treatment was significantly higher than that in the CK (Figure 12A and Appendix A). These functional genes may drive the compositions of bacterial communities; therefore, bacteria further improve the properties of soil. In previous studies, the composition of the bacterial community was found to be driven by functional genes rather than taxonomic or phylogenetic compositions [87,88]. We speculated that these genes promoted the related bacterial growth and development and then formed the bacterial communities in SC-treated soil. The FUNGuild analysis showed the difference in fungal functional genes between SC3 and CK (Figure 12B). The dominant fungal genera in SC3 were *Mortierella*, *Thysanorea*, and *Calyptella* (Figure 8), which were classified as undefined saprotrophs in the FUNGuild analysis. Saprotrophic fungi are the main decomposers of litter, woody debris, and dead roots [89]. However, the functional prediction was inconsistent with the relative abundance analysis. Compared with CK, the “Endophyte-Plant_Pathogen-Wood_Saprotroph” and ”Plant_Pathogen-Soil_Saprotroph-Wood_Saprotroph” guilds were higher in SC3 (Appendix A). This result may have been caused by the alteration in the fungal communities. We speculated that the relative abundance and richness of the saprotroph pathogens did not increase, but the number of the pathogens perhaps increased. The “Animal_Pathogen-Soil_Saprotroph” guild had a significantly higher abundance in the CK. This may illustrate that there were more earthworms in the SC-treated soil than in the CK. However, further studies are required to confirm the relationship between the functional prediction and fungal communities.

## 5. Conclusions

In this study, a nanonetwork-structured soil conditioner was constructed, and its effects on pepper growth, soil properties, and the construction of microbial communities were evaluated. Our results showed that SC could increase the yield of pepper and change the microbial community. The microbial abundances and diversity were higher in the SC treatment than in the CK. These microbes promote plant growth and improve soil quality. The compositions of bacterial genera and fungal genera in the SC3 samples were correlated with AK, AN, pH, and AP. Compared with the CK, the community compositions in SC3 included more metabolism-related bacterial genes. The complex changes in fungal functional genes need to be studied further. This study provides insight into the correlation between network-structured soil conditioners and soil, microorganisms, and plants. This could help us better understand the mechanism by which microorganisms are improved by network-structured SC.

## Figures and Tables

**Figure 1 biology-12-00668-f001:**
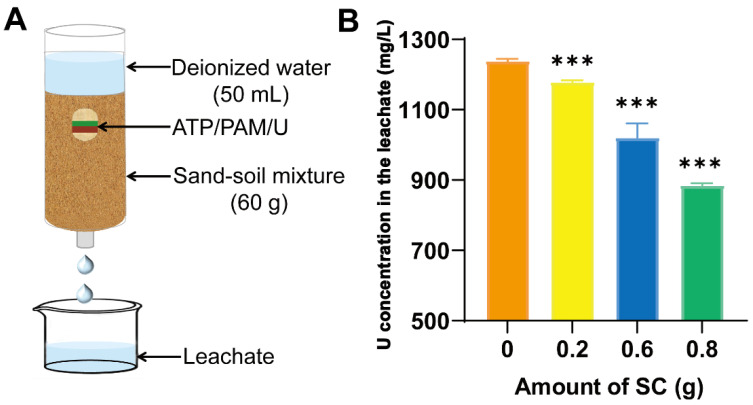
Leaching assay. (**A**) Schematic diagram of the leaching system. (**B**) Influence of the amount of SC on the leaching loss of urea from soil. Notes: ATP: attapulgite; PAM: polyacrylamide; U: urea. Asterisks indicate statistically significant differences, as determined by Student’s *t*-test (*** *p* < 0.001).

**Figure 2 biology-12-00668-f002:**
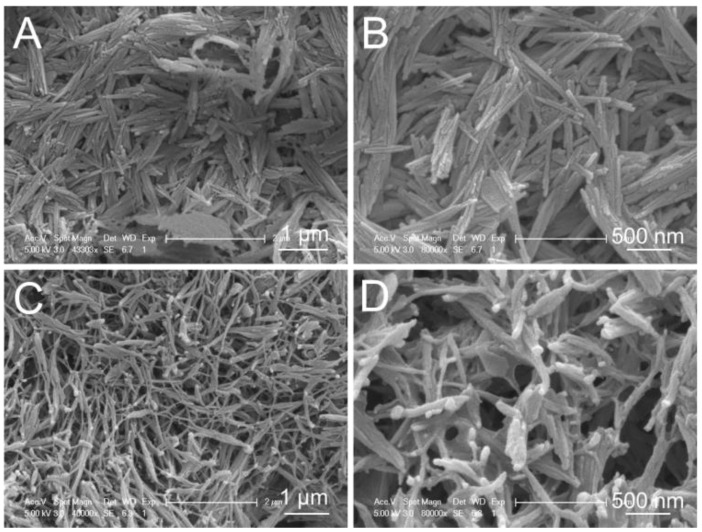
The microstructure of the SC as revealed by SEM with different magnifications: (**A**,**B**) attapulgite; (**C**,**D**) SC.

**Figure 3 biology-12-00668-f003:**
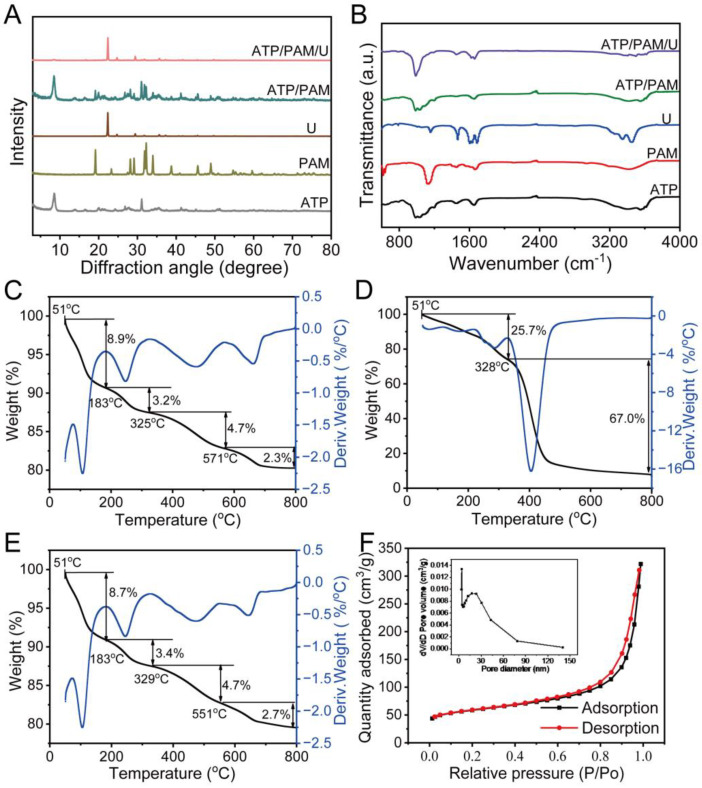
Interaction analyses of the SC system. (**A**) FTIR spectra of attapulgite. (**B**) XRD patterns of attapulgite, polyacrylamide, and SC. (**C**–**E**) TGA (black line) and DTA (blue line) curves of attapulgite, polyacrylamide, and SC. (**F**) N_2_ adsorption–desorption isotherms of SC. (Inset) Pore size distribution of SC. Notes: ATP: attapulgite; PAM: polyacrylamide; U: urea; FTIR: Fourier transform infrared spectrometer; XRD: X-ray diffractometer; TGA: thermal gravimetric analysis; DTA: differential thermal analysis.

**Figure 4 biology-12-00668-f004:**
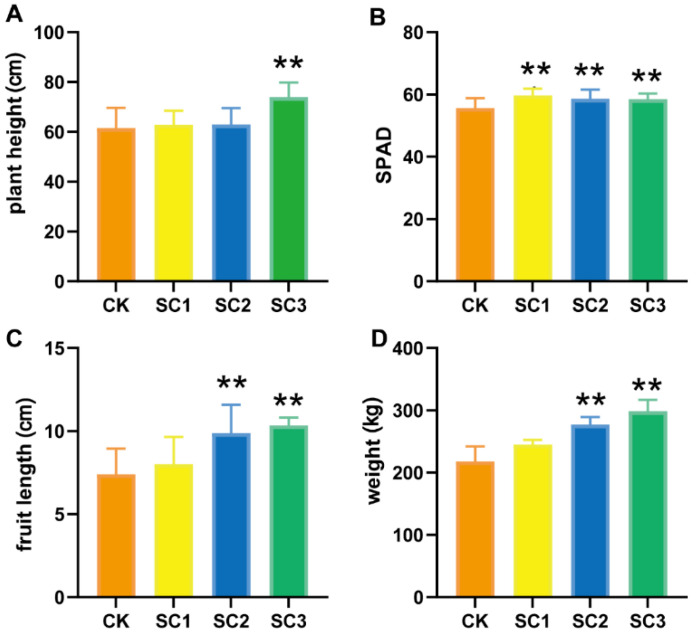
Effects of SC on plant growth. (**A**–**D**) Plant height, SPAD, and fruit length and weight when treated with different amounts of SC. Asterisks indicate statistically significant differences, as determined by Student’s *t*-test (** *p* < 0.01).

**Figure 5 biology-12-00668-f005:**
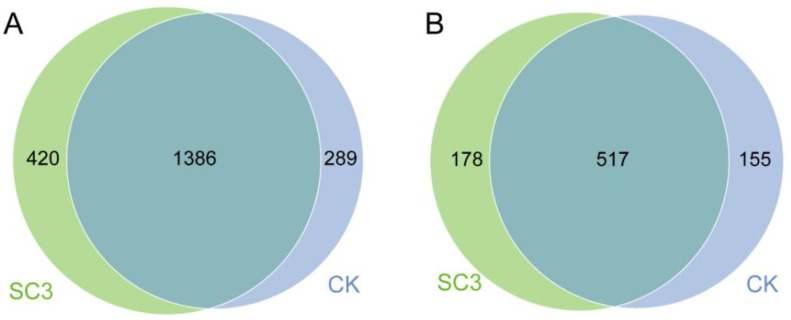
The OTUs between CK and SC3, displayed in a Venn diagram. (**A**) Bacteria, (**B**) fungi. The overlapping part shows the common OTU numbers.

**Figure 6 biology-12-00668-f006:**
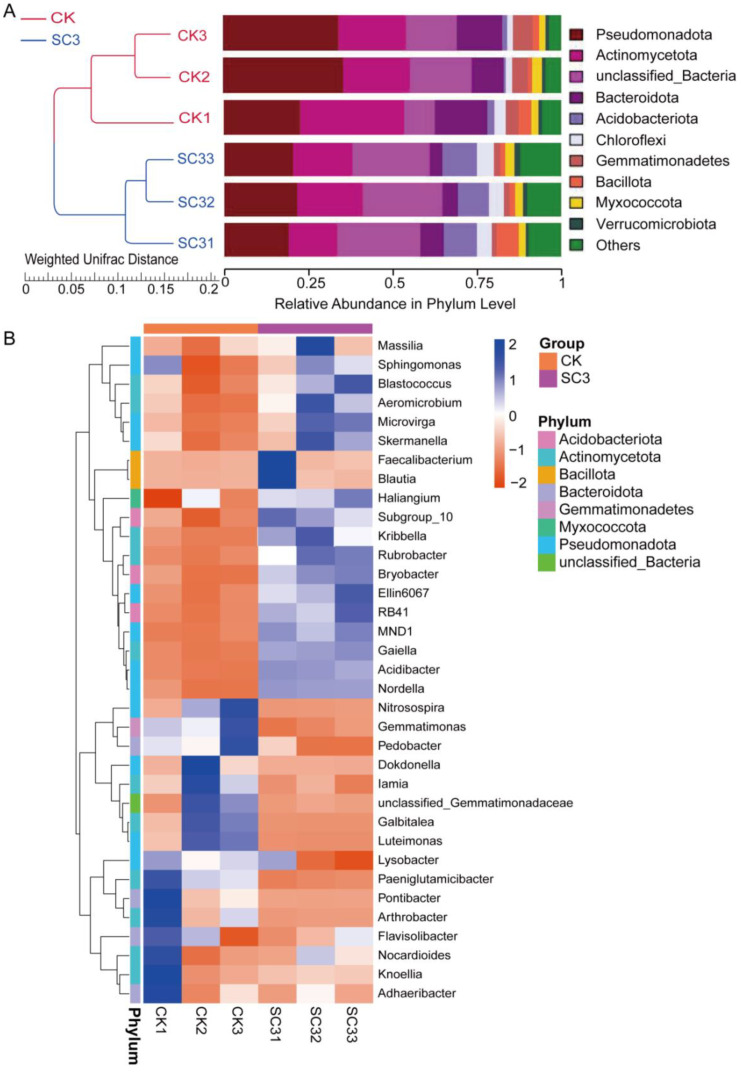
Composition and relative abundance of bacterial communities. (**A**) UPGMA clustering analysis showed the top 10 relative abundance levels of the bacterial community in each sample at the phylum level. (**B**) Clustering heat map of the relative abundance levels of the top 35 bacteria at the genus level in each sample. The “Z” value corresponding to the heat map is described by the color intensity from 2 to −2. The color gradient shifted from blue to red (from low to high abundance). CK1-CK3 represent the 3 replicates of soil without soil conditioner; SC31-SC33 represent the 3 replicates of soil with 400 kg ha^−1^ soil conditioner.

**Figure 7 biology-12-00668-f007:**
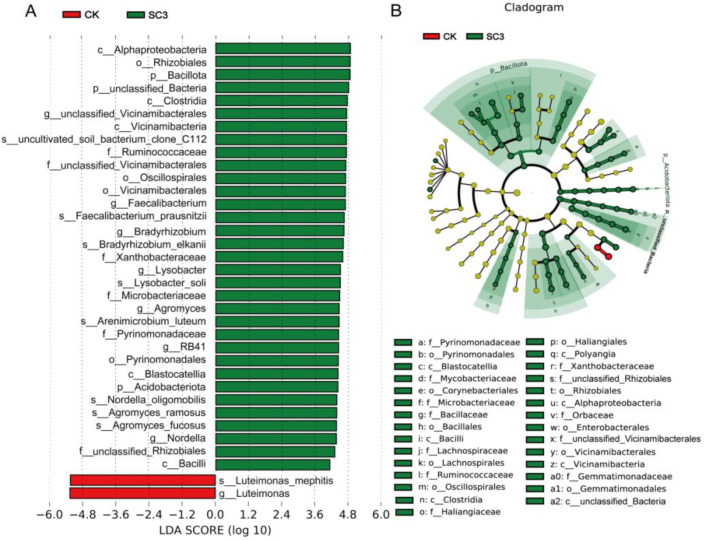
LEfSe analysis of the difference in bacterial abundance between CK and SC3. (**A**) LDA score between CK and SC3. The column length indicates the effect size of the bacterial lineages. (**B**) The cladogram of bacterial communities with differences between CK and SC3. The proportion of bacterial abundance is indicated by the circle’s diameter. Red and green nodes: bacterial taxa that play a vital role in CK and SC3, respectively. Yellow: nonsignificant.

**Figure 8 biology-12-00668-f008:**
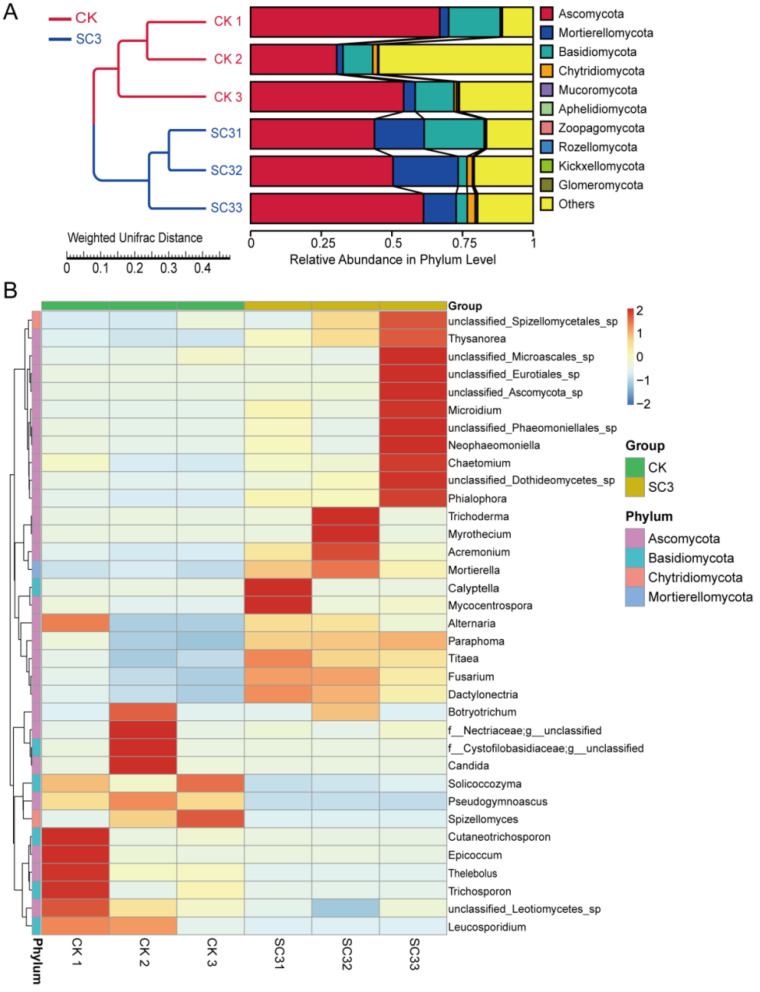
Composition and relative abundance of fungal communities. (**A**) UPGMA clustering analysis: the top 10 relative abundance levels in each sample at the phylum level. (**B**) Clustering heat map of the relative abundance of the top 35 fungi at the genus level in each sample. The corresponding value of the heat map is the “Z” value, depicted by the color intensity, ranging from 2 to −2. The gradient of color shifts from blue (low abundance) to red (high abundance). CK1-CK3 represent the 3 replicates of soil without soil conditioner; SC31-SC33 represent the 3 replicates of soil with 400 kg ha^−1^ soil conditioner.

**Figure 9 biology-12-00668-f009:**
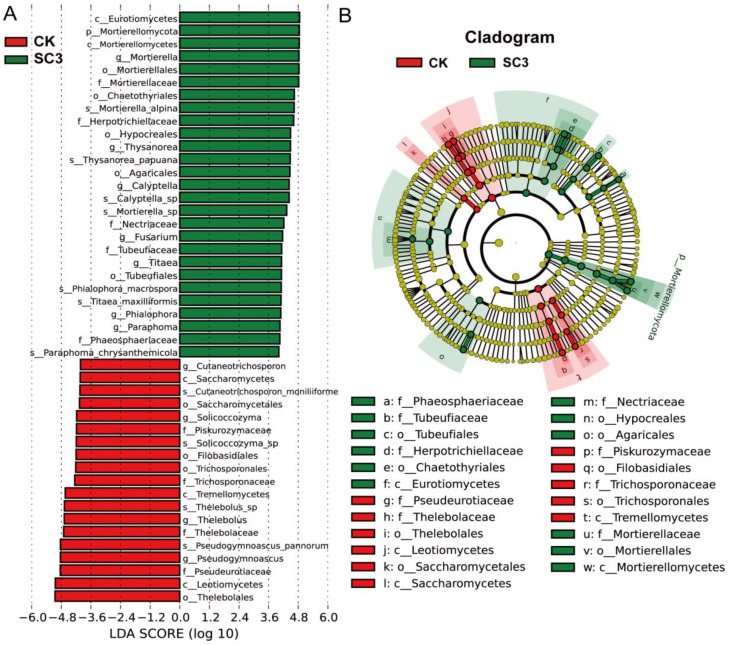
LEfSe analysis of fungal abundance between CK and SC3. (**A**) LDA score between CK and SC3. The column length indicates the effect size of the bacterial lineages. (**B**) The cladogram of fungal communities with differences between CK and SC3. The proportion of bacterial abundance is indicated by the circle’s diameter. Red and green nodes: fungal taxa that play a vital role in CK and SC3, respectively. Yellow: nonsignificant.

**Figure 10 biology-12-00668-f010:**
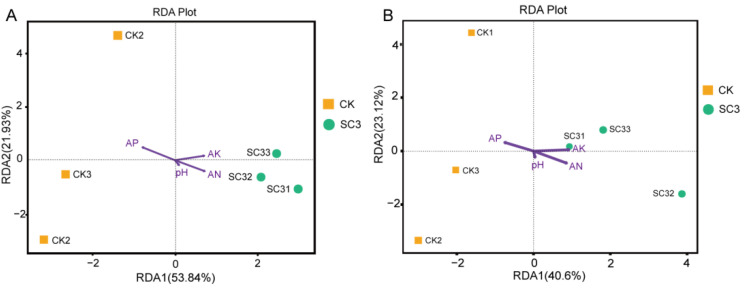
Redundancy analysis (RDA) of microbial genera and soil physical and chemical properties. (**A**,**B**) RDA of bacterial and fungal communities and environmental variables. The arrow length in the RDA plot corresponds to the strength of the correlation between a variable and the community structure. CK1-CK3 represent the 3 replicates of soil without soil conditioner; SC31-SC33 represent the 3 replicates of soil with 400 kg ha^−1^ soil conditioner.

**Figure 11 biology-12-00668-f011:**
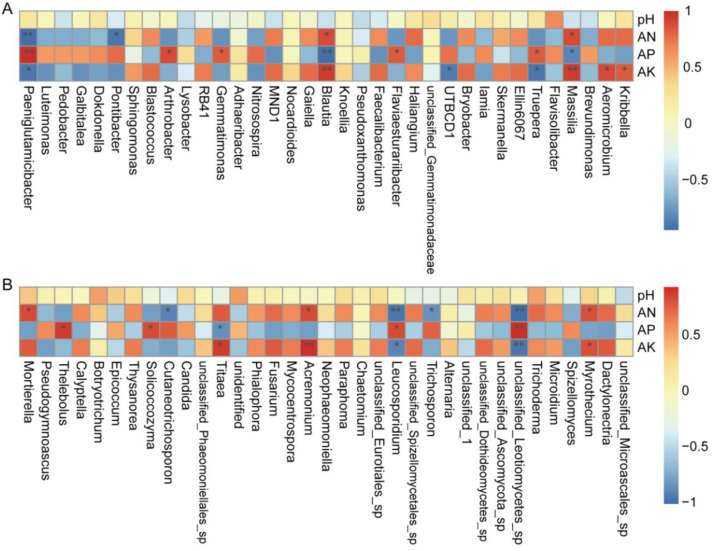
The heatmap of Spearman’s rank correlation. (**A**,**B**) The correlation between the richness of bacterial and fungal genera and environmental factors, where the “r” value is between −1 and 1, r < 0 is a negative correlation, and r > 0 is a positive correlation. Asterisks indicate statistically significant differences, as determined by Student’s *t*-test (* *p* < 0.05, ** *p* < 0.01). The gradient of color shifts from blue to red (from low to high abundance).

**Figure 12 biology-12-00668-f012:**
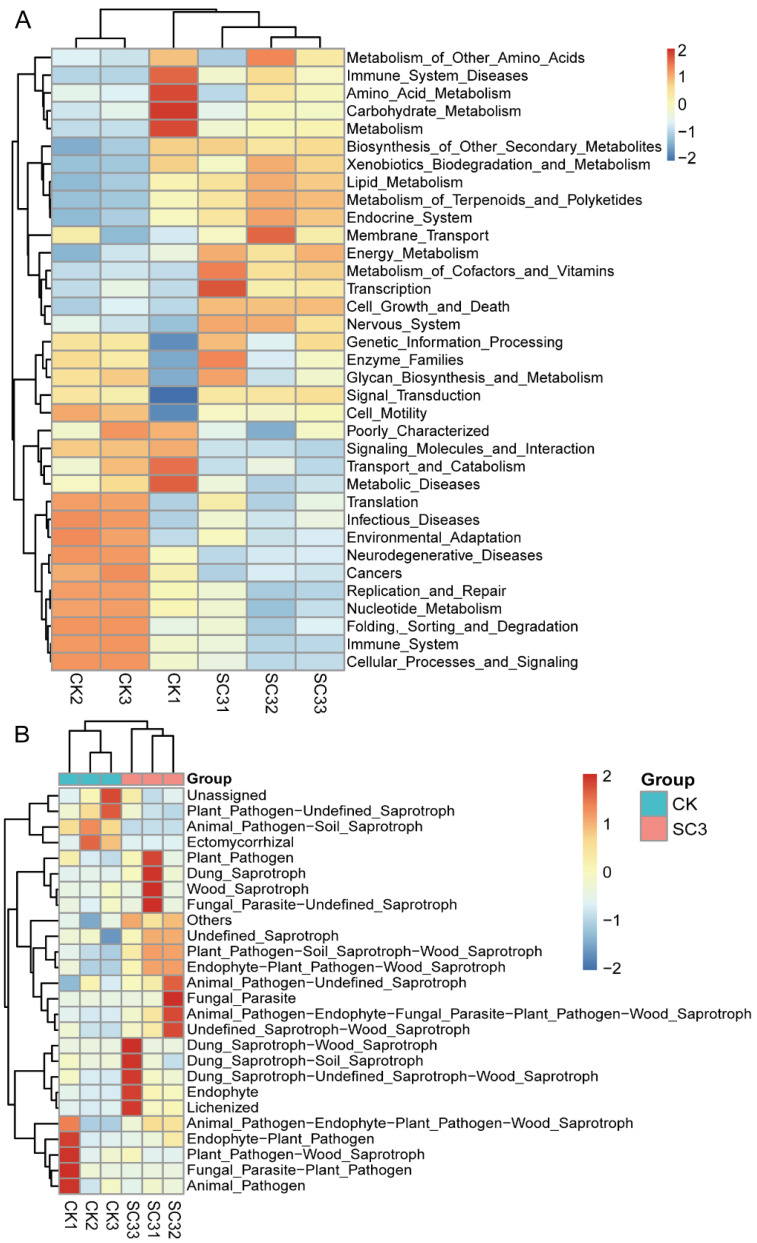
Clustering heat map of the functional pathway analyses. (**A**) The relative abundance of KEGG (level-2) pathways for bacterial communities of all samples. The KEGG pathways are indicated by a gradient of color from blue (low abundance) to red (high abundance). The corresponding value of the heat map is the “Z” value. (**B**) The relative abundance of predicted fungal functions. The corresponding value of the heat map is the “Z” value when the relative abundance was normalized. The gradient of color shifts from blue (low abundance) to red (high abundance). CK1-CK3 represent the 3 replicates of soil without soil conditioner; SC31-SC33 represent the 3 replicates of soil with 400 kg ha^−1^ soil conditioner.

**Table 1 biology-12-00668-t001:** Soil properties after using SC.

Treatment	pH Value	Organic Matter (g/kg)	Total Salt (%)	Available Nitrogen (mg/kg)	Available Phosphorus (mg/kg)	Available Potassium (mg/kg)	Soil Capacity (g/cm^3^)	Porosity (%)	Water Retention Rate (%)
CK	7.86 ± 0.13	26 ± 3	0.10 ± 0.02	80 ± 7	95 ± 2	426 ± 10	1 ± 0.01	17.5 ± 0.34	23 ± 1
SC3	7.87 ± 0.06	26 ± 2	0.11 ± 0.02	97 ± 8	82 ± 4	438 ± 14	1 ± 0.01	19.7 ± 0.97	25 ± 2

The data are shown as the mean ± standard deviation (SD) of three replicates.

**Table 2 biology-12-00668-t002:** The alpha diversity between CK and SC3 treatments.

Treatment		Oberved_Species	chao1	ACE	Shannon	Simpson	PD_Whole_Tree	Good’s_Coverage
16S rRNA	CK	1240 ± 100	1336 ± 101	1346 ± 102	7.01 ± 0.19	0.96 ± 0.00	104 ± 4	0.996
	SC3	1550 ± 8	1614 ± 10 *	1604 ± 18 *	8.85 ± 0.01 ***	0.99 ± 0.00 ***	115 ± 2	0.997
ITS1-5F	CK	435 ± 22	459 ± 21	464 ± 20	4.93 ± 0.67	0.89 ± 0.07	125 ± 6	0.999
	SC3	521 ± 8 **	548 ± 11 **	549 ± 15 **	6.59 ± 0.10 *	0.97 ± 0.00	145 ± 5 **	0.999

The data are shown as the mean ± standard deviation (SD) of three replicates. Asterisks indicate statistically significant differences, as determined by Student’s *t*-test (* *p* < 0.05, ** *p* < 0.01, *** *p* < 0.001).

## Data Availability

The raw read data were submitted to Science Data Bank (https://doi.org/10.57760/sciencedb.07079, accessed on 6 January 2023).

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
