# Peer review of "Effects of a Nanonetwork-Structured Soil Conditioner on Microbial Community Structure"

_biology, 2023, doi:10.3390/biology12050668_

Round 1

Reviewer 1 Report

The manuscript complies with the scope of the Journal. May be published subject to the following considerations:

1.      Line 57 excessive amounts of fertilizers - the light farmer does not use it, because it is not profitable from the economic point of view.

2.      Rows 60-62 - the statement that 50-60% of nitrogen is taken up by crops, and the rest run off into water is imprecise, as it is well known that some nitrogen is also immobilized in the soil.

3.       Line 75 - instead of rhizobium spp. should be Rhizobium spp.

4.      Line 109 - use another abbreviation for attapulgite. The abbreviation ATP is reserved for another chemical compound commonly used in the physiology and biochemistry of organisms. The abbreviation change should be applied consistently throughout the manuscript.

5.      Line 118 - It is questionable whether A 50 g sand-soil mixture is a sufficient mass of soil for this type of research. Soil and sand characteristics are also missing. Maybe it was better to use the soil itself without mixing it with sand. Also, Figure 1 says it was 60g, not 50g. Where did this discrepancy come from?

6.      Chapter 2.3 Greenhouse Experiment and Design - soil characteristics and information about the number of repetitions are missing.

7.      Figures 8 and 10 lack explanation of what the numbers 1, 2 and 3 next to CK and SC3 mean.

8.      The References chapter requires ordering, as it is structured differently, e.g. the title of the journal in position 30 is "Biology and Fertility of Soils", and in position 34 "Environmental microbiology reports". The note applies to the entire chapter.

9. The information in Figure 7 is illegible.Please correct the font size.

Reviewer 2 Report

The authors investigated the effects of a nanonetwork-structured soil conditioner on soil microbial community structure using high-throughput sequencing analysis. Their results are potentially interesting and useful, but I may have the following major and minor comments.

1.      The title is a little long. One suggestion is to show your most interesting/novel results here.

2.      The research background and motivations are not clear from your abstract.

3.      The key result from your abstract is not clear. You need to be very sharp on your key conclusions. It can be clearer if we can report the treatment-induced proportional changes.

4.      There are some unnecessary or uncommonly used abbreviations in your abstract.

5.      Keywords: Better to reduce the overlap with your abstract and title.

6.      Line 39-41. You may need a more informative conclusion and implication here. The current one seems an almighty conclusion. It is unclear what are your main conclusions.

7.      You may also need some specific conclusions and implications based on your key results.

8.      It is unclear from your introduction that why do you want to investigate nanonetwork-structured soil conditioner? Why do you choose high-throughput sequencing analysis? Why are they important? See relevant publikations, Wang et al., 2021, https://doi.org/10.3390/d13090408.

9.      It might be better to have another paragraph highlighting the importance of your study site in the introduction.

10.  We need to pave the way for your research questions, rather than just listing the earlier results. The question or hypothesis should be linked to your main objectives.

11.  There is quite large room to improve the writing. Not only for the writing itself but also for the writing logic.

12.  the research questions or hypotheses are not clear enough from the last paragraph in your introduction.

13.  To make your results comparable, more details on Materials and methods are required for the further evaluation.

14.  More information on the climatic, edaphic, and environmental variables are required.

15.  I was redirected to read your earlier studies while reading your method section. However, this does not really help the reading. We need at least some brief information here.

16.  More information is required on why we choose the sites.

17.  More information on the land use history and plant community composition are required.

18.  Relevant citations are required for your method section. You’d better to cite the original method paper.

19.  Several unnecessary abbreviations are preventing the reading. I have to remember a lot of abbreviations when reading your manuscript.

20.  You need more efforts for the results section to well focus on your key findings. This can make your results clearer. One key result for each paragraph, and then well focused on that key topic.

21.  You need a better logic to link your field and greenhouse experiments.

22.  you need to check data normality before ANOVA.

23.  Some in-depth data analyses and particularly data interpretation are required to explore the underlying mechanisms. Regression analysis may help us understanding the relationships behind. Will you compare the relative importance of these variables? Have you tried to discuss the potential drivers?

24.  To help me understanding your statistical analysis, you may need to briefly mention the purposes of each your statistical analysis.

25.  Some sentences are unnecessary long with changing focuses. It is not easy to understand these long sentences. The writing needs to be improved.

26.  Despite there is no clear rule on the tense, I can find several sentences with inconsistent tense, which need to be double checked.

27.  The results section is a little long. You need to focus on the key/novel results that can serve for your main conclusions.

28.  The main conclusions and key implications are not clear enough from your discussion. The potential mechanisms need to be discussed. What are the important biotic and abiotic factors affecting the responses? What are the key implications? For example, Luo et al., 2022, https://doi.org/10.1111/1365-2435.14178.

29.  Some uncertainties and limitations of your study can be discussed. What are the limitations of your study? What are the future research priorities? Are there some uncertainties for your data analysis and interpretation?

30.  There are usually some “Challenges in upscaling laboratory studies to ecosystems in soil microbiology research”, when you are comparing data from difference sources. This point can be discussed.

31.  The relationships between soil microbes and soil C cycling should be considered.

32.  Correlations do not necessarily mean causal relationships. Cautions are required when you interpret your correlations as causal relationships.

33.  Regarding your references, 1. Missing information can be found for some your references. 2. Be consistent with either upper- or lower-case letters in the title. 3. Pay attention to the author names.

34.  The letters in your figures are not clear enough. Please increase the font size and reduce the abbreviations used in the figures and tables.

35.  Will you show the sample size and standard deviations for each variable in your tables?

36.  You should keep the consistency of decimals across your text, figures and tables, which keeping the validity of your decimals.

Reviewer 3 Report

This study explored a nanonetwork-structured soil conditioner and its impact on soil’s physicochemical properties, rhizosphere microbial communities, and ultimately on crop growth. The research is innovative, and the authors conducted comprehensive analysis from multiple perspectives which combines phenomics and genomics data and provides an understanding of the mechanism of how network-structured nanocomposites affect the soil microenvironment. However, there are several minor issues that need further editing.

Line 145, it’s confusing about the base fertilizer, is it also applied to the control group? If so, please be clear here. Otherwise, we can’t exclude the possibility that the changes in SC group is because of the base fertilizer application.

Line 139 Please specify the latin name of square pepper.

Line 157 could you explain more about how the soil samples were collected according to the S shape?

Line 150 -153 how many seedlings per plot and how are the plots randomized in greenhouse?

Line 166, how many plants per plot were used for collecting each rhizosphere sample? how are they selected?

Line 176 how are primers designed or selected from literature? Please specify and proved references.

Line 186 what are splicing sequences?

Line 189 needs to provide your filtering criteria.

Line 192-195 need to re-write the language to be more scientific.

Line 201 It’s UNITE database.

Line 212 please specify how you did beta diversity (which r package, which function, which kind of distance calculated)

Line 220 “Student’s t-test was used to calculate the differences between the samples.” not sure what data you applied the t-test on. Microbial abundance?

Line 321 could you explain what’s average effectiveness, and how it’s calculated?

Line 356-357 “The relationship of species within the fungal community in the SC3 group was significantly closer than in the CK group.” Could you explain on what evidence this conclusion been made?

Line 371, instead of using self-defined terms like “unidentified_Bacteria”, please use scientific language to describe it. Save as line 352 “observed_species”, “PD_whole_tree”. Line 374 “other phyla”

Line 375, what’s “abundant bacteria”? how to define abundant.

Line 488, if there is no statistical method used, it's not appropriate to say “significantly” here.

Line 613, can you also briefly describe what are these previous studies?

Line 650-652 “The Ascomycota phylum was the most abundant in all of the investigated cropping systems and plays an ecological role as a decomposer.” Need to add references here.
